

# Assessment of different factors on the influence of glass wool concentration for detection of main swine viruses in water samples

Jie Fan[1], Hongjian Chen[1], Wenbo Song[1], Hao Yang[1], Rui Xie[1], Mengfei Zhao[1], Wenqing Wu[1], Zhong Peng[1,2] and Bin Wu[1]

[1] State Key Laboratory of Agricultural Microbiology, The Cooperative Innovation Center for Sustainable Pig Production, College of Veterinary Medicine, Huazhong Agricultural University, Wuhan, China
[2] Hubei Hongshan Laboratory, Wuhan, Hubei, China

## ABSTRACT

Viruses existed in wastewaters might pose a biosecurity risk to human and animal health. However, it is generally difficult to detect viruses in wastewater directly as they usually occur in low numbers in water. Therefore, processing large volumes of water to concentrate viruses in a much smaller final volume for detection is necessary. Glass wool has been recognized as an effective material to concentrate multiple in water, and in this study, we assessed the use of glass wools on concentrating pseudorabies virus (PRV), African swine fever virus (ASFV), and porcine epidemic diarrhea virus (PEDV) in water samples. The influence of pH values, water matrix, water volume, filtration rate, temperature on the effect of the method concentrating these viruses for detection was evaluated in laboratory. Our results revealed that glass wool was suitable for the concentration of above-mentioned viruses from different water samples, and demonstrated a good application effect for water with pH between 6.0–9.0. Furthermore, glass wool also showed a good recovery effect on concentrating viral nucleic acids and viral particles, as well as living viruses. In addition, combining use of glass wool with skim milk, polyethylene glycol (PEG)-NaCl, or ultracentrifuge had good effects on concentrating ASFV, PRV, and PEDV. Detection of wastewater samples ($n = 70$) collected from 70 pig farms in 13 regions across Hubei Province in Central China after glass-wool-concentration determined one sample positive for ASFV, eighteen samples positive for PRV, but no sample positive for PEDV. However, these positive samples were detected to be negative before glass wool enrichment was implemented. Our results suggest that glass wool-based water concentration method developed in this study represents an effective tool for detecting viruses in wastewater.

Corresponding author
Bin Wu, wub@mail.hzau.edu.cn

## INTRODUCTION

Human domestic sewages and livestock feces discharging into environments have been recognized as a main reason for viral contamination in water (*Owa, 2013*). It has been discovered that more than 100 different types of viruses are discharged through human and animal feces (*Fong & Lipp, 2005*). The presence of these viruses in wastewater may pose a severe risk to human and animal health (*Abd-Elmaksoud et al., 2014*). Therefore, all wastewater is required to be treated to meet specific standards before discharging. However, it is difficult to eliminate all viral agents from wastewater through regular treatments, even when chlorination disinfection is use (*Adefisoye et al., 2016*; *Fioretti et al., 2017*; *Naidoo & Olaniran, 2013*; *Wong, Onan & Xagoraraki, 2010*). In this regard, wastewater has been recognized as a key biosecurity risk point for monitoring in both medical and veterinary activities (*Bogler et al., 2020*).

In general, the concentration of viruses in wastewater is low and direct virus detection is frequently ineffective (*Blanco et al., 2019*). Therefore, a large volume of water should be concentrated into a smaller volume to increase the virus concentration, allowing the next step for virus detection (*Ikner, Gerba & Bright, 2012*). The surface of glass wool coated with mineral oil has hydrophobic and positive potential points, making it adsorb virus with negative charged in pH-neutral water and then elute without adding additional reagents (*Lambertini et al., 2008*). In addition, the cost of glass wool is low, and glass wool it is suitable for concentrating large volume of water with low threshold for equipment, thereby representing a promising material for concentrating viruses in water (*Abd-Elmaksoud et al., 2014*; *Blanco et al., 2019*; *Powell et al., 2000*).

China is the largest pig farming country in the world, and the pig industry plays an important role in China's agriculture and economy (*Wu et al., 2020*). However, prevalence and occurrence of swine diseases, particularly several types of viral diseases, including African swine fever (ASF), porcine epidemic diarrhea (PED), pseudorabies (PR), and porcine reproductive and respiratory syndrome (PRRS), pose severe threats to the development of pig industry in China (*Liu et al., 2021*; *Su et al., 2020*; *You et al., 2021*). After the spread of ASF into China in 2018 (*Zhou et al., 2018*), Chinese pig farms improved their biosecurity construction with strict controls on incoming personnel, materials, vehicles, pigs, media, and feed, attempting to eliminate viruses present or carried in each part (*Dixon, Sun & Roberts, 2019*). It has been also widely recognized that water used in pig farms, including drinking water, as well as wastewater discharging from both pigs and farm workers, may pose a severe risk point for disease prevention and biosecurity construction (*Dixon et al., 2020*). However, it is still lack of effective methods to concentrate water for detecting pathogenic agents, making it difficult to assess the risk. In this study, we developed a method of water concentration using glass wool for ASF virus (ASFV), PR virus (PRV), and PED virus (PEDV) detection. Using this method, we performed an investigation of these three viruses in water samples collected from 70 farms in 13 regions across Hubei, an important pig rearing and pork producing in China.

## MATERIALS AND METHODS

### Virus strains

Different types of viruses were used for evaluating the effects of the water-concentration method developed in this study. Among these types of viruses, ASFV strain HuB-2 was isolated from the lung of a pig, and the evaluation based on ASFV was performed in the Animal Biosafety Level III Laboratory of Huazhong Agricultural University. Considering the biosecurity risk of using wildtype viral strains, we selected two attenuated vaccine strains for the assessment of the method on concentrating PRV (strain HB98; Keqian Bio., Wuhan, China) and PEDV (strain AJ1002; Keqian Bio., Wuhan, China). In addition, our previously collected *Salmonella* bacteriophage ph2-2 (*Zhao et al., 2022*) was also included for evaluation in this study. Initial viral solutions containing ASFV (188,456 copies/μl), PRV (291,288 copies/μl), and PEDV (322,130 copies/μl) were prepared.

### Preparation of glass wool

Preparation of glass wool was optimized based on previous studies (*Kiulia et al., 2010*; *Lambertini et al., 2008*; *Millen et al., 2012*) to reduce the amount of water and processing reagents used. Briefly, glass wool (U-1339; Johns Manville, Denver, CO, USA) was soaked in double-distilled water for 15 min, then soaked in 0.5 M hydrochloric acid for 20 min, rinsed three times with double-distilled water, and soaked in 0.5 M sodium hydroxide for 20 min, then rinsed three times with double-distilled water. The processed glass wool was packed into filters, which were circular PVC containers with a diameter of 45 mm and a length of 105 mm, with interfaces on both sides to connect with pipelines. Approximately 60 g of dry processed glass wool were placed in each filter, and finally, the glass wool was stored in PBS (pH 6.7–7.0) at 4 °C for future use.

### Primary concentration method

Before the experiment, the equipment is wiped and operating table surface with 0.5% sodium hypochlorite solution (*Abd-Elmaksoud et al., 2014*), and then wipe with water after 15 min. Use a peristaltic pump (YZ1515X; Runze, Shenzhen City, China) to extract the seeded virus water from the container and filter it through a glass wool filter element at different speeds, and let the peristaltic pump continue to work for 3 min after all the water has been filtered. Soak the glass wool filter element with 75 ml of 3% beef extract buffer (B8570; Solarbio, Beijing, China) solution containing 0.5 M glycine (1275KG2P5; BioFroxx, Hangzhou, China) and pH 9.0 for 20 min, then wash with 75 ml of beef extract buffer solution again. Collect the total 150 ml buffer solution in a clean container, adjust the eluent pH to neutral with 1M hydrochloric acid, and store at 4 °C; if over 48 h, it must be stored at −20 °C or lower temperature.

### Assessment of the influence of different factors on primary concentration of PRV, ASFV and PEDV

To assess the influence of different factors on viral enrichment by primary concentration, a series water samples containing ASFV, PRV, and PEDV under the following conditions were prepared for primary concentration. Viral nucleic acids in water samples before and

after concentration were detected using real-time fluorescence quantitative PCR (qPCR), and the results were compared.

(1) **pH values:** To test the influence of pH values on virus recovery, PBS (4,000 ml) containing different types of viruses (200 µl) at different pH values (6.0, 7.0, 8.0, 9.0) were prepared since the pH values of environmental waters generally ranged from 6.0 to 9.0.

(2) **Water types:** Water samples were prepared by seeding viruses (200 µl) in samples (4,000 ml) collected from different sources, including collected from different sources, including tap water (pH = 8.0, nephelometric turbidity unit (NTU) = 0, containing no organic matters and low concentrations of salt ions), urban inland lake water (pH = 9.0, NTU = 17, containing rich in organic matters and microorganisms), water from the mainstream of Yangtze River (pH = 7.9, NTU = 9.0, containing silts), water from suburban rivers (pH = 7.8, NTU = 25.0, receiving some domestic sewages from nearby villages), and PBS (pH = 7.4, NTU = 0, containing salt ions but no organic matters).

(3) **Filtration speeds:** PBS solutions (pH 7.4; 4,000 ml) mixed with 200 µl of different viruses were filtered at 500, 1,000, and 1,500 ml/min; at the same time, PBS solution without mixed virus was set as a negative control group.

(4) **Filtration volumes:** Three groups of PBS solutions (pH 7.4) solutions were set, with volumes of 4,000, 12,000, and 20,000 ml respectively; a certain volume of virus mixed solution was added (ensuring the initial virus concentration of the sample before filtration was consistent) and mixed evenly; at the same time, PBS solution without mixed virus was set as a negative control group.

(5) **Temperatures:** Three groups of PBS solutions were set, and the temperature of PBS was adjusted to 4 °C, 20 °C and 32 °C respectively. Then 200 µl of virus mixed solution was added to each group and mixed evenly; at the same time, a negative control group without mixed virus was set for each temperature.

## Effects of glass wool enrichment on viral particles and nucleic acids

To access the effect of glass wool on the enrichment of viral particles and nucleic acids, three types of virus-associated-samples were prepared: (1) "viral particles"; this type of sample was prepared by removing free viral nucleic acids thoroughly through addition of Benzanase (Merck, Darmstadt, Germany) and incubated at 37 °C for 20 min (*Berg et al., 2016*). (2) "nucleic acid"; this type of sample was prepared by extracting viral nucleic acids using either a commercial viral DNA or RNA preparation kit (Vazyme, Nanjing, China). (3) "viral solution"; no treatment was given and there might be both viral particles and nucleic acids inside. Thereafter, each of the prepared samples was added into 4,000 ml PBS for primary concentration. Finally, viral DNA/RNA in the concentration products of virus-associated-samples were extracted and were quantified by qPCR.

## Evaluation of the efficacy of glass wool on concentrating live virus

To assess the efficacy of glass wool on enriching living virus, two types of phage-associated samples were prepared: (1) "phage particles" which was prepared by removing the free nucleic acids using Benzanase (Merck, Darmstadt, Germany); and (2) "phage solution" ($1 \times 10^{11}$ PFU/ml for live phages or for phage nucleic acids) did not receive any special

treatment. After that, either phage particles or phage solutions (200 μl) were speeded into 4,000 ml PBS for primary concentration by glass wool. The concentration products of phage particles were incubated with its host bacterium (*Salmonella* Paratyphi strain 201107 (*Zhao et al., 2022*)) for titering, while the DNAs in the concentration products of phage solutions were quantified by qPCR.

## Secondary concentration effectiveness evaluation

A total of 150 ml of negative beef extract powder eluents containing 200 μl of ASFV, PRV, PEDV, or *Salmonella* bacteriophage ph2-2 were for secondary concentration, respectively. Three methods were used for secondary concentration: skimmed milk method, PEG-NaCl method and ultracentrifugation. The skimmed milk method was to add 0.2‰ skimmed milk (CN7861; Coolaber, Beijing, China) powder to 40 ml of eluent and adjust the pH to 3.5. Shake it at 200 ×g for 2 h at room temperature and centrifuge it at 2,000 ×g for 30 min. The precipitate was resuspended in 0.01 M PBS and stored at −80 °C (*Assis et al., 2017*; *Calgua et al., 2008*). The PEG-NaCl method was to add 15% PEG8000 (1363GR; BioFroxx, Hangzhou, China) and 0.2 M NaCl to 40 ml of eluent. Shake it at 200 ×g for 2 h at 4 °C after PEG is dissolved. Stand it still at 4 °C overnight; the next day, eluent was centrifuged at 4,500 ×g for 45 min and the precipitate was resuspended in 0.01 M PBS and stored at −80 °C (*Abd-Elmaksoud et al., 2014*; *Lambertini et al., 2008*). The ultracentrifugation method was to add 5 ml of 30% sucrose to 30 ml of eluent, then centrifuged at 30,000 ×g and 4 °C for 2 h. The precipitation is redissolved with 0.01 M PBS and stored at −80 °C (*Ammersbach & Bienzle, 2011*).

## Concentration of wastewaters collected from pig farms and detection of different types of viruses

Wastewater samples collected from 70 pig farms in 13 regions of Hubei Province (including Wuhan, Xiangyang, Yichang, Xiaogan, Huanggang, Xianning, Shiyan, Enshi, Jingmen, Jingzhou, Huangshi & Ezhou, Tianmen & Qianjiang & Xiantao, and Suizhou) between June 1 and December 31, 2021 were concentrated by the method developed in this study for detecting the contamination of ASFV, PRV, and PEDV. In each region, 4–6 commercial pig farms were selected. Due to the requirements of biosecurity control in pig farms, all samples were collected by farm workers and delivered to the pig farm wall, then they were carried to laboratory for primary concentration within 48 h after collection. Following this, secondary concentrations were conducted for qPCR detection of ASFV, PRV, and PEDV.

## qPCR assays

Total DNAs and/or RNAs were extracted from 200 μL water samples. RNAs were transcribed into cDNA by a reverse transcription reagent kit (RRA036, Takara, Japan) immediately. The detection of viral nucleic acids was performed using qPCR (CF96X; Bio-rad, Hercules, CA, USA), following the primers (Table S1) and protocols described previously (*Chen et al., 2023*; *Guo et al., 2016*; *Lin et al., 2020*). The standard curve was established as follows: a slightly longer gene sequence than the target fragment was

designed, usually around 300–600 bp, and a specific primer was designed for PCR amplification and Gel recovery. The recovered fragment was introduced into the pMD-19T vector in DH5α cells. After single colony identification and sequencing through PCR, successful colonies were selected for further cultivation and then the plasmid was extracted and the $OD_{260}$ was read to calculate the plasmid copy number concentration. Then, the copy number was diluted to $10^{-7}$ by $10^{-1}$, $10^{-2}$, $10^{-3}$, and the Ct value was detected by qPCR method for each gradient. The relationship between the Ct value and copy number was established and a standard curve was plotted.

### Bacteriophage titering

Bacteriophage was titered as described previously (Zhao et al., 2022). Briefly, *Salmonella* phage ph2-2 was diluted with PBS (pH 7.4) from $10^{-1}$, $10^{-2}$, $10^{-3}$, to $10^{-7}$, a total of seven gradients or estimated gradients. Next, 300 μL of *Salmonella* bacterial solution was cultured for 12 h and 1 ml of diluted ph2-2 bacteriophage virus solution were mixed with 7 ml of melted 45 °C semi-solid agar medium. The mixed semi-solid medium was quickly poured into a prepared TSA agar plate and gently rotated on a laminar flow hood to distribute evenly. The agar was allowed to solidify and then incubated overnight at 37 °C. Afterwards, the transparent plaques were observed and counted.

### Statistical analysis

Data were analyzed statistically using Prism software 8.0 (GraphPad, San Diego, CA, USA) and expressed as the mean ± the standard errors (SE). Comparisons among different groups were evaluated using multiple t-tests-one per row. For Figs. 1–3: $*p \leq 0.05$, statistically different; $**p \leq 0.01$, highly statistically different; $***p \leq 0.001$, significantly statistically different.

## RESULTS

### The influence of different factors on the enrichment of PRV, ASFV, and PEDV by glass wool

Overall, ASFV, PRV, and PEDV could be enriched by glass wool effectively at these pH values, and the highest recovery rate was observed for the enrichment of PRV, followed by ASFV and PEDV, respectively (Fig. 1A). No statistical difference was observed for the enrichment of PRV and ASFV in water samples with different pH values (Fig. 1A).

In different types of water samples, the recovery rates of PRV, ASFV, and PEDV ranged between 10.9% and 80.4% (tap water, 10.9%; water from urban inland lakes, 67.9%; water from Yangtze River, 49.5%; water from suburban rivers, 80.4%; PBS, 36.9%), 17.6% and 48.4% (tap water, 28.7%; water from urban inland lakes, 17.6%; water from Yangtze River, 43.2%; water from suburban rivers, 48.4%; PBS, 29.5%), 4.9% and 6.9% (tap water, 6.9%; water from urban inland lakes, 4.9%; water from Yangtze River, 5.9%; water from suburban rivers, 5.1%; PBS, 6.8%), respectively (Fig. 1B). Strikingly, different types of water samples posed a significant influence on the enrichment of PRV and ASFV (Figs. 1B and 1C). The viral recovery rates in water samples with volumes of 4,000, 12,000, and 20,000 ml were 30.6%, 38.9%, and 27.8% respectively for ASFV, and 59.1%, 53.0% and 45.9%

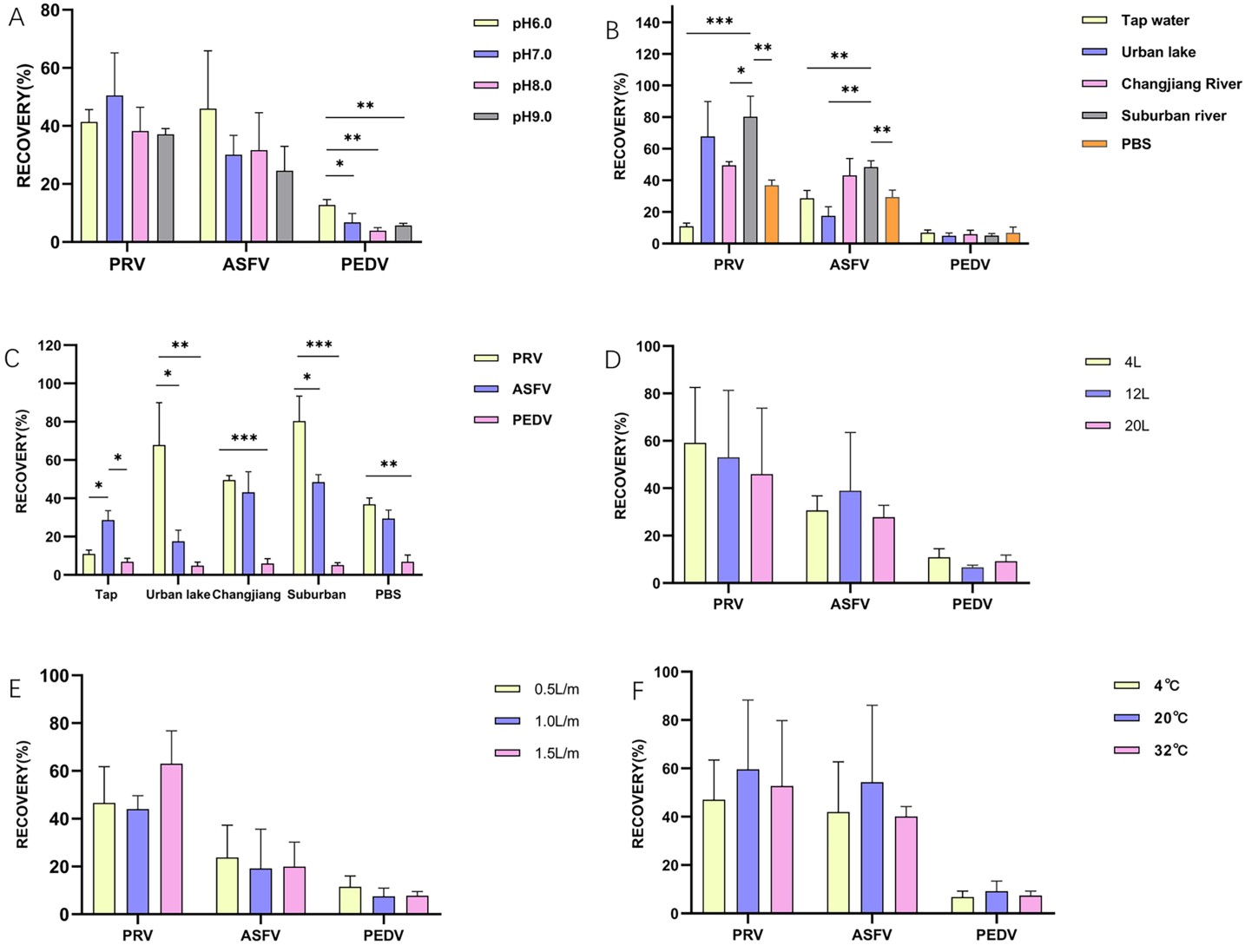

**Figure 1 The influence of different factors on the enrichment of PRV, ASFV, and PEDV by glass wool.** (A & B) The recovery rates of different viruses concentrated by glass wool from water samples with different pH values and/or different water matrixes, respectively; (C) the recovery rates of different viruses in each type of water samples concentrated by glass wool; (D–F): the recovery rates of different viruses concentrated by glass wool from water samples with different volumes (D) or at different filtration speeds (E) or at under different temperatures (F). $^{*}p \leq 0.05$, statistically different; $^{**}p \leq 0.01$, highly statistically different; $^{***}p \leq 0.001$, significantly statistically different.

respectively for PRV, as well as 10.9%, 6.2% and 9.2% respectively for PEDV. There was no difference on recovery rates of the three viruses from different water volumes (Fig. 1D).

Regarding the influence of different speeds for filtration (500, 1,000, 1,500 ml/min), no difference was observed on recovery rates of the three viruses, but the overall trend was increased slightly as the filtering speed increased (Fig. 1E). The recovery rates of PRV under above-mentioned speeds were 45%, 50.8%, and 58.9%, respectively; while those for ASFV were 13.5%, 20.2%, 21.0%, and 8.6%, 8.0%, 8.1% for PEDV, respectively. Next, we investigated the influence of different temperatures, the results revealed that although higher recovery rates were observed for the enrichment of the three viruses at 20 °C than

those for the three viruses at 4 °C and 32 °C (PRV: 59.6% (20 °C) *vs*. 47.0% (4 °C) *vs*. 52.7% (32 °C); ASFV: 54.3% (20 °C) *vs*. 42.0% (4 °C) *vs*. 40.1% (32 °C); PEDV: 9.2% (20 °C) *vs*. 6.8% (4 °C) *vs*. 7.4% (32 °C), but no statistical difference was observed between the viral enrichment under different temperatures (Fig. 1F).

## The efficacy of glass wool on concentrating viral particles and nucleic acids

Considering wastewater may harbor different types of virus-associated agents, including viral particles, viral nucleic acids, or viral particles plus nucleic acids released by dead viruses (marked as viral solutions), we therefore assessed the efficacy of glass wool on concentrating the above-mentioned different types of virus-associated agents. The results revealed that the recovery rates of virus solutions, viral particles, and nucleic acids in PBS of PRV were 70.8%, 55.4%, and 44.8%, respectively (Fig. 2A). For ASFV, the recovery rates for the three types of virus-associated agents in PBS were 28.3%, 24.9%, and 39.7%, respectively (Fig. 2A). However, those for PEDV were 3.9%, 3.5%, and 18.1%, respectively (Fig. 2A).

The efficacy of glass wool on concentrating viral particles and nucleic acids was also evaluated using *Salmonella* phage as a model. In PBS containing phage particles, the recovery rate quantified by phage titering was 8.0%, while that quantified by qPCR was 17.3% (Fig. 2B). The recovery rate yielded by glass wool concentrating live phages and that yielded by glass wool concentrating phage DNAs exhibited a statistical difference ($P < 0.01$). In PBS containing phage solutions, the recovery rate quantified by qPCR was 10.9%, which was lower than that of the phage particle group (17.3%, $P < 0.05$) (Fig. 2B).

## The efficacy of different methods for secondary concentration

According to the quantify results achieved by qPCR detecting viral nucleic acids, a highest recovery rate of PRV was found in second concentration using the skimmed milk method (56%), followed by the PEG-NaCl method (30.1%) and the ultracentrifugation method (3.48%). The skimmed milk method had significantly higher concentration efficiency than the other two methods (Fig. 3A). However, the highest recovery rate of ASFV was observed in the PEG method (27.8%), followed by the skimmed milk method (15.4%) and the ultracentrifugation method (6.68%). The PEG-NaCl method also had significantly higher concentration efficiency than the other two methods (Fig. 3A). For secondary concentration of PEDV, a significantly higher recovery rate was observed in the ultracentrifugation method (20.9%) and the PEG-NaCl method (19.9%) compared to that in and the skimmed milk method (10.6%) (Fig. 3A). *Salmonella* phage was also used to assess the role of different methods for secondary concentration on living viruses. The results demonstrated that second concentration through both ultracentrifugation method (79.97 ± 32.27%) and PEG-NaCl method (45.85 ± 29.49%) recovered living viruses, but almost no living phages were recovered by the skimmed milk method (Fig. 3B).

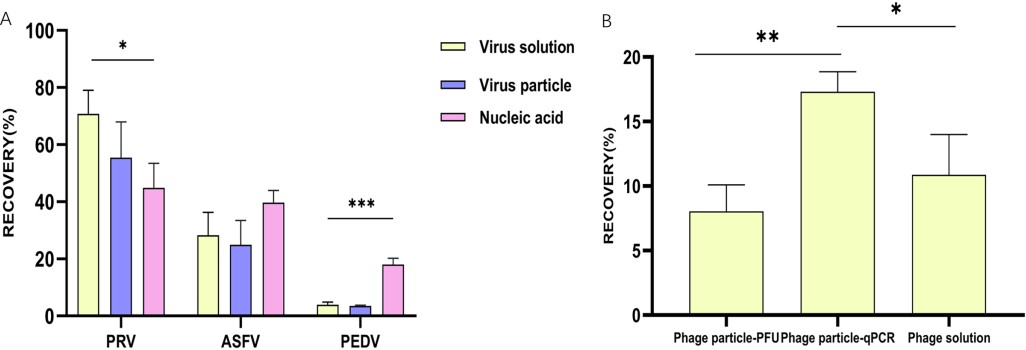

**Figure 2  The efficacy of glass wool on concentrating viral particles and nucleic acids.** (A) A column chart showing the recovery rates of glass wool concentrating particles and nucleic acids of different viruses; (B) a column chart showing the recovery rates of live phages and nucleic acids concentrated by glass wool from water samples. $^*p \leq 0.05$, statistically different; $^{**}p \leq 0.01$, highly statistically different; $^{***}p \leq 0.001$, significantly statistically different.

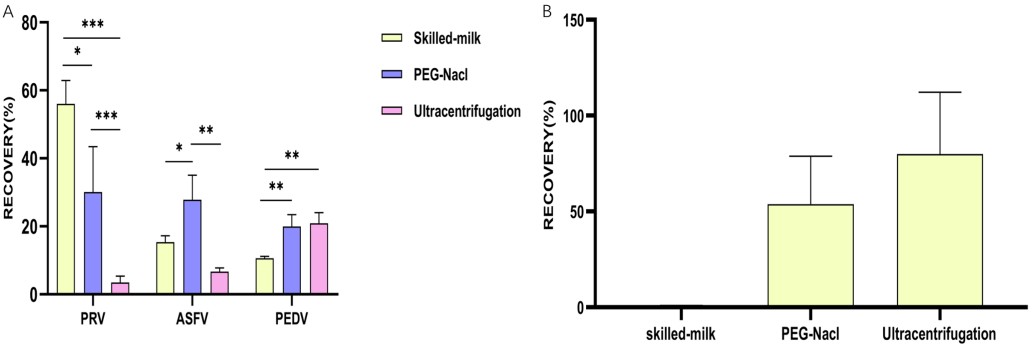

**Figure 3  Evaluation of different methods for secondary concentration.** (A) A column chart showing the recovery rates of different viruses achieved by different methods for secondary concentration; (B) a column chart showing the recovery rates of *Salmonella* phages achieved by different methods for secondary concentration. $^*p \leq 0.05$, statistically different; $^{**}p \leq 0.01$, highly statistically different; $^{***}p \leq 0.001$, significantly statistically different.

## Detection of ASFV, PRV, and PEDV in farm wastewater

To assess the contamination of ASFV, PRV, and PEDV in pig farm wastewater, water samples collected from 70 farms in Hubei Province were adjusted the pH values to range in 6.0–9.0, and set for primary and secondary concentrations followed by qPCR detecting the target agents. Among the 70 samples, only one sample (1.43%, 1/70) collected from a farm in Xiangyang was detected to be positive for ASFV (Table 1). The Ct value for this sample was 35.12. However, this sample was detected as a negative one before concentration. In addition, 18 samples (25.7%, 18/70) were detected to be positive for the gH gene of PRV but negative for the gE gene of PRV, suggesting PRV detected in these samples were vaccine strains. In contrast only one sample (Sample NO.66, CT value 37.4) was positive for PRV gH gene before concentration. Strikingly, all 70 samples were concentrated and detected to be negative for PEDV.

**Table 1 Sampling in pig farms and city rivers in Hubei province.**

| Region | Cities | Farm numbers | No. of pigs farmed | Farm types | Water types | Volume (L) | Virus detection | | |
|---|---|---|---|---|---|---|---|---|---|
| | | | | | | | ASFV | PRV-gH | PEDV |
| North Hubei | Shiyan | 1 | 200 | Sow farm | RW | 10 | N/A | N/A | N/A |
| | Shiyan | 2 | 500 | Sow farm | RW | 5 | N/A | N/A | N/A |
| | Shiyan | 3 | 800 | Sow farm | RW | 10 | N/A | N/A | N/A |
| | Shiyan | 4 | 2,000 | Sow farm | BS | 10 | N/A | N/A | N/A |
| | Shiyan | 5 | 1,000 | Sow farm | IW | 10 | N/A | 35.8 | N/A |
| | Suizhou | 6 | 20,000 | Fattening farm | RW | 10 | N/A | N/A | N/A |
| | Suizhou | 7 | 8,000 | Sow farm | IW | 20 | N/A | 39.21 | N/A |
| | Suizhou | 8 | 8,000 | Sow farm | IW | 20 | N/A | 36.05 | N/A |
| | Xiangyang | 9 | 3,000 | Sow farm | BS | 10 | N/A | N/A | N/A |
| | Xiangyang | 10 | 2,000 | Sow farm | IW | 10 | N/A | N/A | N/A |
| | Xiangyang | 11 | 2,000 | Sow farm | IW | 10 | N/A | 37.17 | N/A |
| | Xiangyang | 12 | 6,000 | Sow farm | IW | 5 | N/A | N/A | N/A |
| | Xiangyang | 13 | 3,000 | Fattening farm | IW | 10 | N/A | N/A | N/A |
| | Xiangyang | 14 | 6,000 | Sow farm | IW | 20 | 35.12 | 30.05 | N/A |
| East Hubei | Ezhou | 15 | 500 | Sow farm | IW | 15 | N/A | N/A | N/A |
| | Huanggang | 16 | 200 | Sow farm | RW | 10 | N/A | 35.41 | N/A |
| | Huanggang | 17 | 7,000 | Sow farm | IW | 20 | N/A | N/A | N/A |
| | Huanggang | 18 | 2,000 | Sow farm | IW | 10 | N/A | N/A | N/A |
| | Huanggang | 19 | 3,000 | Sow farm | IW | 10 | N/A | N/A | N/A |
| | Huanggang | 20 | 800 | Sow farm | RW | 5 | N/A | N/A | N/A |
| | Huangshi | 21 | 600 | Sow farm | BS | 10 | N/A | N/A | N/A |
| | Huangshi | 22 | 800 | Sow farm | BS | 10 | N/A | N/A | N/A |
| | Huangshi | 23 | 5,000 | Sow farm | RW | 10 | N/A | N/A | N/A |
| | Wuhan | 24 | 5,000 | Sow farm | IW | 10 | N/A | N/A | N/A |
| | Wuhan | 25 | 10,000 | Fattening farm | RW | 10 | N/A | N/A | N/A |
| | Wuhan | 26 | 20,000 | Fattening farm | IW | 10 | N/A | N/A | N/A |
| | Wuhan | 27 | 7,000 | Sow farm | IW | 15 | N/A | 36.64 | N/A |
| | Wuhan | 28 | 1,500 | Sow farm | RW | 20 | N/A | 36.69 | N/A |
| | Wuhan | 29 | 10,000 | Fattening farm | RW | 10 | N/A | N/A | N/A |
| | Wuhan | 30 | 5,000 | Sow farm | IW | 5 | N/A | N/A | N/A |
| | Wuhan | 31 | 5,000 | Sow farm | IW | 10 | N/A | 34.51 | N/A |
| | Wuhan | 32 | 600 | Sow farm | IW | 5 | N/A | N/A | N/A |
| | Xiaogan | 33 | 20,000 | Fattening farm | RW | 3 | N/A | N/A | N/A |
| | Xiaogan | 34 | 6,000 | Sow farm | IW | 15 | N/A | N/A | N/A |
| | Xiaogan | 35 | 800 | Sow farm | IW | 10 | N/A | 34.23 | N/A |
| | Xiaogan | 36 | 5,000 | Sow farm | IW | 15 | N/A | N/A | N/A |
| | Xiaogan | 37 | 5,000 | Sow farm | IW | 15 | N/A | 37.28 | N/A |
| | Xiaogan | 38 | 20,000 | Fattening farm | RW | 10 | N/A | N/A | N/A |

| Region | Cities | Farm numbers | No. of pigs farmed | Farm types | Water types | Volume (L) | Virus detection | | |
|---|---|---|---|---|---|---|---|---|---|
| | | | | | | | ASFV | PRV-gH | PEDV |
| West Hubei | Enshi | 39 | 500 | Sow farm | RW | 10 | N/A | N/A | N/A |
| | Enshi | 40 | 500 | Sow farm | RW | 10 | N/A | N/A | N/A |
| | Enshi | 41 | 400 | Sow farm | RW | 5 | N/A | N/A | N/A |
| | Enshi | 42 | 200 | Sow farm | RW | 10 | N/A | N/A | N/A |
| | Yichang | 43 | 1,000 | Sow farm | RW | 6 | N/A | N/A | N/A |
| | Yichang | 44 | 3,000 | Fattening farm | IW | 20 | N/A | N/A | N/A |
| | Yichang | 45 | 500 | Sow farm | IW | 5 | N/A | N/A | N/A |
| | Yichang | 46 | 2,000 | Sow farm | IW | 10 | N/A | N/A | N/A |
| | Yichang | 47 | 3,000 | Sow farm | IW | 30 | N/A | N/A | N/A |
| | Yichang | 48 | 1,000 | Sow farm | BS | 10 | N/A | N/A | N/A |
| South Hubei | Jianghan | 49 | 500 | Sow farm | RW | 5 | N/A | N/A | N/A |
| | Jianghan | 50 | 600 | Sow farm | RW | 10 | N/A | N/A | N/A |
| | Jianghan | 51 | 600 | Sow farm | RW | 10 | N/A | N/A | N/A |
| | Jianghan | 52 | 3,000 | Sow farm | IW | 10 | N/A | 37.69 | N/A |
| | Jianghan | 53 | 3,000 | Sow farm | IW | 10 | N/A | 34.48 | N/A |
| | Jingmen | 54 | 1,000 | Sow farm | BS | 10 | N/A | N/A | N/A |
| | Jingmen | 55 | 500 | Sow farm | IW | 10 | N/A | N/A | N/A |
| | Jingmen | 56 | 600 | Sow farm | IW | 10 | N/A | N/A | N/A |
| | Jingmen | 57 | 1,000 | Sow farm | RW | 10 | N/A | N/A | N/A |
| | Jingmen | 58 | 2,000 | Sow farm | IW | 10 | N/A | N/A | N/A |
| | Jingmen | 59 | 500 | Sow farm | IW | 10 | N/A | N/A | N/A |
| | Jingmen | 60 | 300 | Sow farm | RW | 10 | N/A | N/A | N/A |
| | Jingzhou | 61 | 3,000 | Sow farm | IW | 10 | N/A | N/A | N/A |
| | Jingzhou | 62 | 3,000 | Sow farm | IW | 10 | N/A | N/A | N/A |
| | Jingzhou | 63 | 5,000 | Sow farm | IW | 10 | N/A | 37.05 | N/A |
| | Jingzhou | 64 | 5,000 | Sow farm | IW | 10 | N/A | 35.72 | N/A |
| | Xianning | 65 | 2,000 | Fattening farm | IW | 10 | N/A | N/A | N/A |
| | Xianning | 66 | 2,000 | Sow farm | IW | 10 | N/A | 31.89 | N/A |
| | Xianning | 67 | 3,000 | Sow farm | IW | 10 | N/A | 37.51 | N/A |
| | Xianning | 68 | 8,000 | Sow farm | BS | 10 | N/A | N/A | N/A |
| | Xianning | 69 | 7,000 | Sow farm | IW | 10 | N/A | 31.53 | N/A |
| | Xianning | 70 | 3,000 | Sow farm | IW | 10 | N/A | N/A | N/A |

**Note:**

Jianghan is region including Tianmen & Qianjiang & Xiantao; RW is raw water, IW is irrigation water, BS is biogas slurry; N/A is detected negative by qPCR.

The environmental protection facilities of pig farms are different, and the sewage treatment degree is different. Raw water is the supernatant of the feces; irrigation water refers to the water that reaches the irrigation standard after sewage treatment, which is clear and neutral pH; biogas slurry is the supernatant after fecal fermentation.

## DISCUSSION

In this study, we assessed the application of glass wool for primary concentration of different types of porcine viruses in wastewater samples. It has been reported that glass wool is a preferred low-cost material for concentrating viruses in water samples

particularly samples with large volumes with multiple-notable advantages (*Blanco et al., 2019*; *Mabasa et al., 2022*; *Sedji et al., 2018*). Concentrating viruses in water samples using glass wool does not require the adjustment of pH values lower than 7.5, or without adding metal ions, which makes the concentration more convenient (*Blanco et al., 2017*; *Pérez-Sautu et al., 2012*). Correspondingly, adjusting the pH of large volume water samples is a difficult operation, and wastewater pH is often higher than 7.5. Based on these reasons, we focused on evaluating the pH effect and found no significant difference in the concentration efficiency of ASFV and PRV when water pH values ranged between 6.0–9.0, greatly improving the application prospects of the glass wool method. In addition, virus type and water matrix are important factors that affect the concentration efficiency of glass wool, which is consistent with the results of other studies (*Lambertini et al., 2008*). Here we also demonstrated that the recovery rates given by glass wool on concentrating ASFV, PRV and PEDV in the same water samples, as well as on concentrating specific virus in different types of water samples were quite different. Therefore, specific optimizations should be given on glass wool-based concentration of specific viral species in specific water matrix.

In this study, the recovery rate of PEDV in water samples using glass wool was relatively lower than those of ASFV and PRV concentration, which is suggestive of a various efficacy of glass wool on concentrating different types of viruses. It should be noted that the positively charged glass wool mainly absorbs negatively discharged enveloped or non-enveloped viruses from large volume of water samples through covalent binding (*Blanco et al., 2019*, *2017*). Therefore, viral biochemical properties may affect the concentration efficacy of glass wool. However, this influence might be partly counteracted by optimizing several factors associated with the concentration. For example, a recent study increases the recovery rate of transmissible gastroenteritis virus (TGEV) in large volumes of water samples (from 0.4% to 5.1%) by adjusting the pH value of eluents, as well as taking the other measures such as increasing the concentration of PEG (*Blanco et al., 2019*). Therefore, glass wool-based method might be displayed inclusiveness and flexibility when concentrating different types of viruses. For example, according to the results provided in this study, there is no need to improve the existing methods for the concentration of ASFV, but the method should be optimized if PEDV is the primary concern.

In different strategies applied for secondary concentration, a higher recovery rate of PRV and ASFV was calculated by qPCR method when using skimmed milk method, but the recovery rate was nearly zero when determining the infectious recovery. It is conjectured that skimmed milk powder can simultaneously adsorb positively charged virus particles and nucleic acid fragments in an acidic environment, but the virus will rapidly inactivate at pH 3.5, causing a significant decrease in infectivity. The PEG-NaCl method mainly relies on intermolecular force compression to cause virus precipitation and has better effects on RNA viruses, which is similar to the results of a previous study (*Amdiouni et al., 2012*). Ultracentrifugation generates high centrifugal force to precipitate virus particles. The differences in the concentration principles of the three methods result in differences in the recovery results of infectivity. The skimmed milk method in the secondary concentration results is the simplest operation, with the least amount of reagent

loss and the most obvious precipitation, and a good concentration effect. Ultracentrifugation can obtain more live viruses. Clinically, the nucleic acid can be first detected using skimmed milk powder, and then live viruses can be obtained using Ultracentrifugation or PEG method, which is both convenient and fast.

In this study, a low rate was observed for the detection of ASFV in wastewater samples collected from pig farms Hubei Province. This finding suggests a good control of ASF in pig farms in Hubei, which may be owing to the great success achieved by Chinese pig farms taking strict biosecurity actions (48–72 h personnel isolation, disinfection of materials, vehicle drying, high temperature granulation for feed) to prevent and control the disease. As a World Organisation for Animal Health (WOAH) listed disease, ASF is highly contagious among domestic and wild pigs with a mortality rate can reach 100% (*Dixon et al., 2020*). However, there is still no effective drugs or vaccines commercially available for this contagious disease. Therefore, Chinese pig farms have taken strict biosecurity actions to control the disease since its occurrence in 2018 and those actions have achieved a great success (*Liu et al., 2021*). While 25.7% (18/70) of the samples were detected to be positive for PRV, all of them were characterized as vaccine-strain as they were negative for the gE gene. This data might suggest a good result achieved by pseudorabies eradication actions in Chinese pig farms in recent years (*Xia et al., 2018*). It is a bit surprising for the un-detection of PEDV. A possible reason to explain this result may be that wastewaters have received a series of strict treatments before their discharging (*e.g.*, adding disinfectants, setting for fermentation, long-term storage) and these treatments may lead to the degradation of PEDV RNA, which is usually not stable in environment.

To be concluded, we assessed the use of glass wool for concentrating ASFV, PRV and PEDV in water samples in this study. Our results demonstrated that glass wool was a good choice for large volume water concentration for detecting ASFV and PRV, but different factors, particularly water matrix, may affect the recovery efficacy. Therefore, specific optimizations should be given on glass wool-based concentration of specific viral species in specific water matrix. Detection of important porcine viruses in pig farm wastewater is also a useful method to assess the biosafety of pig farms.

## ACKNOWLEDGEMENTS

We sincerely acknowledge staffs at pig farms for wastewater sample collection. We also thank Prof. Huanchun Chen at Huazhong Agricultural University for his advices on revising the manuscript.

## ABBREVIATIONS

| | |
|---|---|
| **ASF** | African swine fever |
| **ASFV** | African swine fever virus |
| **NTU** | Nephelometric turbidity unit |
| **PEG** | Polyethylene glycol |
| **PED** | Porcine epidemic diarrhea |
| **PEDV** | Porcine epidemic diarrhea virus |
| **PR** | Pseudorabies |

| PRV | Pseudorabies virus |
| PRRS | Porcine reproductive and respiratory syndrome |

### Funding

This work was supported by the National Natural Science Foundation of China (Grant No. U20A2059), the Hubei Provincial Key Research and Development Program (Grant No. 2021BBA085), the Yingzi Tech & Huazhong Agricultural University Intelligent Research Institute of Food Health (No. IRIFH202209), the Modern Agricultural Industrial Technology System of Hubei Province (No. HBHZD-ZB-2020-005), and the Startup Fund from Hubei Hongshan Laboratory & Huazhong Agricultural University. The funders had no role in study design, data collection and analysis, decision to publish, or preparation of the manuscript.

### Grant Disclosures

The following grant information was disclosed by the authors:
National Natural Science Foundation of China: U20A2059.
Hubei Provincial Key Research and Development Program: 2021BBA085.
Yingzi Tech & Huazhong Agricultural University Intelligent Research Institute of Food Health: IRIFH202209.
Modern Agricultural Industrial Technology System of Hubei Province: HBHZD-ZB-2020-005.
Hubei Hongshan Laboratory & Huazhong Agricultural University.

### Competing Interests

The authors declare that they have no competing interests.

### Author Contributions

- Jie Fan conceived and designed the experiments, performed the experiments, analyzed the data, prepared figures and/or tables, authored or reviewed drafts of the article, and approved the final draft.
- Hongjian Chen performed the experiments, analyzed the data, prepared figures and/or tables, and approved the final draft.
- Wenbo Song performed the experiments, analyzed the data, prepared figures and/or tables, and approved the final draft.
- Hao Yang performed the experiments, prepared figures and/or tables, and approved the final draft.
- Rui Xie performed the experiments, prepared figures and/or tables, and approved the final draft.
- Mengfei Zhao performed the experiments, prepared figures and/or tables, and approved the final draft.

- Wenqing Wu performed the experiments, prepared figures and/or tables, and approved the final draft.
- Zhong Peng conceived and designed the experiments, authored or reviewed drafts of the article, and approved the final draft.
- Bin Wu conceived and designed the experiments, authored or reviewed drafts of the article, and approved the final draft.

## Data Availability

The raw data are available in the Supplemental Files.

## Supplemental Information

Supplemental information for this article can be found online at http://dx.doi.org/10.7717/peerj.16171#supplemental-information.

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
