# Peer review of "Assessment of different factors on the influence of glass wool concentration for detection of main swine viruses in water samples"

_PeerJ, doi:10.7717/peerj.16171_

## Round 0.1 · original submission · Minor Revisions

Please revise the manuscript according to the reviewers' comments.

Reviewer 1 ·

Basic reporting

This study provides valuable insights into the use of glass wool as an effective and practical method for concentrating viruses in wastewater. The findings demonstrate its reliability in detecting specific viruses, highlighting its potential for biosecurity monitoring and public health measures.

Experimental design

It is well-designed and well-supported using strong data.

Validity of the findings

The findings are very straightforward and informative supported by solid data.

Additional comments

Great job done by all the authors!

Annotated reviews are not available for download in order to protect the identity of reviewers who chose to remain anonymous.

Reviewer 2 ·

Basic reporting

Fan et al. reported the investigation of glass wool concentration for detecting swine viruses in a series of different water samples. Overall, the manuscript is well written. The presentation of experimental data is clear. The reference is in good consistency. I will recommend acceptance after a few minor problems are solved.

Experimental design

no comment.

Validity of the findings

The data are robust and clear. The authors should clearly define the recovery of the enrichment experiments for readers to better understand the results. The figure legend of Fig 1A did not mention the pH effect.

The figure legend are too repetitive.

in Line 226-227, the recovery rate in different water sample is very different. The authors should explain more about the underlying reason, instead of stating the data.

In the detection section, the authors claimed that the positive rate of ASFV and RPV are very low and suggest the strict actions in controling the disease in Chinese pig farms. I can see that the recovery of PEDV is low in data presented in Fig 1 and 2, Can it be also due to the relatively low recovery rate of concentration methods proposed by the authors?

Reviewer 3 ·

Basic reporting

Line 97 to 105 could be reformatted with the experimental material being the subject of the sentence. For example, "wipe the equipment" should be "the equipment is wiped".

The images in Figure 1 to 3 should have higher resolution. Besides, color-coding of the bars, or more distinguishable features are recommended for different experimental groups.

A paragraph with all the abbreviations and their full names should be included for a better understanding of the experimental procedures.

Experimental design

No comment.

Validity of the findings

No comment.

---

## Round 0.2 · accepted · Accept

Congratulations! Your paper is accepted.

Reviewer 1 ·

Basic reporting

This study provides valuable insights into the use of glass wool as an effective and practical method for concentrating viruses in wastewater. The findings demonstrate its reliability in detecting specific viruses, highlighting its potential for biosecurity monitoring and public health measures.

Experimental design

It is well-designed and well-supported using strong data.

Validity of the findings

The findings are very straightforward and informative supported by solid data.

Additional comments

Great job done by all the authors!

Reviewer 2 ·

Basic reporting

The authors have addressed the questions properly. The manuscript can be accepted now.

Experimental design

no comment

Validity of the findings

no comment